# Beneficial Effects of *Limosilactobacillus fermentum* in the DCA Experimental Model of Irritable Bowel Syndrome in Rats

**DOI:** 10.3390/nu15010024

**Published:** 2022-12-21

**Authors:** María Jesús Rodríguez-Sojo, Jorge Garcia-Garcia, Antonio Jesús Ruiz-Malagón, Patricia Diez-Echave, Laura Hidalgo-García, José Alberto Molina-Tijeras, Elena González-Lozano, Laura López-Escanez, María Elena Rodríguez-Cabezas, Maria José Rodríguez-Sánchez, Alba Rodríguez-Nogales, Cristina Mediavilla, Julio Galvez

**Affiliations:** 1Departamento de Farmacología, Centro de Investigación Biomédica (CIBM), Universidad de Granada, 18071 Granada, Spain; 2Instituto de Investigación Biosanitaria de Granada (ibs.GRANADA), 18071 Granada, Spain; 3Servicio de Digestivo, Hospital Universitario Virgen de las Nieves, 18071 Granada, Spain; 4Department of Psychobiology, Mind, Brain, and Behavior Research Center (CIMCYC), University of Granada, 18071 Granada, Spain; 5Centro de Investigación Biomédica en Red de Enfermedades Hepáticas y Digestivas (CIBEREHD), 08036 Barcelona, Spain

**Keywords:** IBS rat model, probiotic, *Limosilactobacillus fermentum CECT5716*, Intestine anti-inflammatory activity, visceral analgesia

## Abstract

*Limosilactobacillus fermentum CECT5716*, a probiotic strain isolated from human milk, has reported beneficial effects on different gastrointestinal disorders. Moreover, it has shown its ability to restore altered immune responses, in association with microbiome modulation in different pathological conditions. Therefore, our aim was to assess the effects of a *Limosilacbacillus fermentum CECT5716* in a rat experimental model of irritable bowel syndrome (IBS) that resembles human IBS. The experimental IBS was induced by deoxycholic acid (DCA) in rats and then, *Limosilactobacillus fermentum CECT5716* (10^9^ CFU/day/rat) was administered. Behavioral studies, hyperalgesia and intestinal hypersensitivity determinations were performed and the impact of the probiotic on the inflammatory and intestinal barrier integrity was evaluated. Additionally, the gut microbiota composition was analyzed. *Limosilactobacillus fermentum CECT5716* attenuated the anxiety-like behavior as well as the visceral hypersensitivity and referred pain. Moreover, this probiotic ameliorated the gut inflammatory status, re-establishing the altered intestinal permeability, reducing the mast cell degranulation and re-establishing the gut dysbiosis in experimental IBS. Therefore, our results suggest a potential use of *Limosilactobacillus fermentum CECT5716* in clinical practice for the management of IBS patients.

## 1. Introduction

Irritable bowel syndrome (IBS) is a highly prevalent disorder, in which a dysfunction in the gut–brain axis seems to have a prominent role. This gut disorder is characterized by abdominal discomfort, together with pain and altered bowel habits, associated stomach bloating and flatulence [1]. It is estimated that 1 in 10 people worldwide suffers from IBS [2], but its prevalence differs across reporting regions considering the disparity of the various potential risk factors, including diet, gastrointestinal infections, genetics, and gut microbiome composition [3]. The aetiology of IBS remains not completely unveiled; however, recent experimental and clinical studies have shown that both physiological and psychological variables are responsible for triggering the IBS symptoms [4]. The pathophysiological mechanisms comprise alterations in the intestinal epithelial barrier function, in the composition of the gut microbiota (termed as dysbiosis) and in the immune response, characterized by an increased number of intraepithelial lymphocytes, as well as enteroendocrine and mast cells, in the gut mucosa [5]. In addition, psychological factors, such as anxiety, stress, or depression, have also been shown to bias symptom perception [6], although, at present, it is unknown if they are the cause or the consequence of the development of IBS [7]. Furthermore, one of the main manifestations observed in IBS is intestinal hypersensitivity in association with impairment of the intestinal function, which often coexists with other frequent symptoms, including loss of appetite, sleep disturbances, stress, and depressive states [8,9].

Therefore, the complexity of the pathophysiological mechanisms involved in IBS together with a multifactorial etiopathogenesis makes its therapeutic management a challenge [10], which involves, in most cases, the combination of lifestyle and dietary interventions with behavioral and pharmacological strategies. In this regard, many different types of drugs have been proposed to exert beneficial effects in this condition, including antidepressants, laxatives, prokinetics, antidiarrheal drugs, serotonin 5-HT3 receptor antagonists, antibiotics, intestinal secretagogues, opiate derivatives, stimulants of gabapentin-mediated neuronal signaling as well as natural products [11,12] although their efficacy is controversial [13]. In consequence, nowadays no pharmacological treatment that combines optimal efficacy and safety has been reported. However, recent studies have described the modulatory effects of some nutraceuticals on mast cells which may influence (increase or reduce) IBS symptoms [5]. Moreover, since IBS patients display an altered gut microbiome, its selective modulation with using probiotics and/or prebiotics has emerged as a promising therapy. In this sense, *Limosilactobacillus fermentum CECT5716* is a probiotic strain isolated from human milk that has shown to prevent the development of mastitis in breastfeeding women [14], to reduce gastrointestinal infections in infants [15], as well as to modulate the gut dysbiosis and the altered intestinal barrier function in experimental rodent models of colitis [16,17], obesity [18] and hypertension [19], among others.

Considering all the above, the objective of the present study was to assess the beneficial effects of *L. fermentum* in a rat model of chronic post-inflammatory visceral pain induced by deoxycholic acid (DCA), which produces persistent visceral hyperalgesia and referred pain resembling human IBS. Additionally, the impact of *L. fermentum* administration has been evaluated on behavioral alterations, specifically those associated with stress, together with its capacity to modulate the altered immune response and/or restore the dysbiosis described in this experimental model of IBS. The effect of the probiotic treatment in this IBS experimental model in rats was compared with two treatments used in IBS patients; gabapentin (a GABAergic agent) and rifaximin (an antibiotic) that have been clinically assayed for visceral pain, diarrhea and abdominal discomfort in human IBS [20,21,22].

## 2. Materials and Methods

### 2.1. Reagents, Drugs and Probiotics

The chemicals and drugs were purchased from Sigma Chemical (Madrid, Spain), except when mentioned specifically. The probiotic *Limosilactobacillus fermentum CECT5716* was provided by Biosearch, S.A. (Granada, Spain) and grown in MRS media for a period of 24 h at 37 °C under anaerobic conditions using the Anaerogen system (Oxoid, Basingstoke, UK). For probiotic treatment, bacteria were daily suspended in sterile phosphate-buffered saline (PBS) solution.

The in vitro studies was performed in complete DMEM Advanced (Gibco, ThermoFisher Scientific, Waltham, MS, USA) (containing 10% fetal bovine serum, 2 mM glutamine (Lonza Barcelona, Spain), 1% penicillin/streptomycin and 1% amphotericin B) and cultured at 37 °C and 5% CO_2_.

### 2.2. Cell Viability and Proliferation Assay

Viability and proliferation of rat basophilic leukemia-2H3 basophils (RBL-2H3) obtained from the Cell Culture Unit of the University of Granada (Granada, ES, Spain) and human mast cells (HMC 1.2; Sigma Aldrich, MO, USA) after Compound 48/80 (50 μg/mL), *L. fermentum* (1 × 10^9^ CFU/mL), gabapentin (5, 10 or 25 μM) or rifaximin (5, 10 or 25 μM) incubation were assessed by CellTiter 96^®^ AQueous One Solution Cell Proliferation Assay (MTS) from Promega (Madison, WI, USA) following the recommended protocol. Concisely, the cells were seeded into 96-well plates and after 24 h, the [3-(4,5-dimethylthiazol-2-yl)-5-(3-carboxymethoxyphenyl)-2-(4-sulfophenyl)-2H-tetrazolium] (MTS) solution was added to each well and incubated for 1–4 h. Then, the absorbance of the supernatant was measured at 490 nm. Cell viability and proliferation (%) were calculated by comparing sample absorbance values with untreated control cultures. 

### 2.3. Quantification of β-Hexosaminidase Release In Vitro

The cell degranulation response was quantified measuring the level of β-hexosaminidase released in culture supernatants from RBL-2H3 and HMC 1.2.

RBL-2H3 (1.25 × 10^6^ cells/mL) and HMC 1.2 (1.25 × 10^6^ cells/mL) cells were incubated in 96-well plates for 24 h at 37 °C under 5% CO_2_. Next day, after washing, cells were incubated for 30 min with *L. fermentum* (1 × 10^9^ CFU/mL), gabapentin or rifaximin at 5, 10 or 25 μM diluted in Tyrode’s buffer (137 mM NaCl, 2.7 mM KCl, 1 mM MgCl_2_, 1.8 mM CaCl_2_, 0.2 mM Na_2_HPO_4_, 12 mM NaHCO_3_, 5.5 mM D-Glucose). Then, cells were stimulated with Compound 48/80 at 50 μg/mL. Tyrode’s buffer and Compound 48/80 at 50 μg/mL diluted in Tyrode’s buffer were used as negative and positive controls, respectively. After 90 min of incubation with the substrate solution (3.5 mg/mL p-nitrophenyl-N-acetyl-β-D-glucosaminide in 40 mM citric acid, pH 4.5) at 37 °C, the supernatants were collected and the β-hexosaminidase released was measured. The reaction was stopped by adding glycine 400 mM, pH 10.7, and optical density was measured at 405 nm in a microplate reader (Tecan, Männedorf, CH, Switzerland). The spontaneous enzyme release was deducted from this total value and data were expressed as a percentage (%) of the total β-hexosaminidase content in the cells measured by lysing cells with 0.1% Triton X100. 

### 2.4. Rat Model of Irritable Bowel Syndrome by DCA

The animal studies were performed in agreement with the “Guide for the Care and Use of Laboratory Animals” as promulgated by the National Institute of Health, ARRIVE guidelines and the protocols approved by the Ethics Committee of Laboratory Animals of the University of Granada (17/07/2020/084). Adult (10 weeks old) male Sprague Dawley rats (240–320 g) were purchased from Janvier Labs (St Berthevin Cedex, FR) and housed in Makrolon cages (5 rats/cage; 2 cages per group), maintained with a 12 h light–dark cycle and provided with ad libitum food and tap water in an air-conditioned atmosphere (20–22 °C and 45–55% humidity). The rats were randomly assigned to 5 study groups (*n* = 10). To develop IBS, rats were fasted overnight. Next day, rats were anesthetized and 1 mL of 4 mmol/L DCA in Krebs’ solution was administered through the colon by a gavage needle once daily for three consecutive days. Rats were kept on a mound of bedding in a head-down position to avoid leakage of DCA. A control group (Non-IBS) was included for reference, which was given the DCA vehicle. Once IBS had developed, the different compounds were administered daily by oral gavage, for 14 days. Rats were treated with rifaximin (150 mg/kg), gabapentin (70 mg/kg) or *L. fermentum* (10^9^ CFU/day) depending on the group. The IBS control group followed the same protocol but received sterile water (vehicle used for all compounds). Rats’ body weights, water and food intake and diarrhea episodes were daily recorded during the experiment by a blinded observer. After finishing the administration and physical/behavioral determinations, rats were sacrificed with an overdose of halothane. Then, the colon was removed aseptically and kept frozen at −80 °C or fixed in 10% neutral buffered formalin until future studies.

### 2.5. Measurement of Response to Colorectal Distension

Visceral hypersensitivity was evaluated 7 and 14 days after the last administration of DCA by the response of rats to colorectal distension (CRD) as previously described [23]. Briefly, under light anesthesia, a mini balloon was trans-anally inserted into the colon and inflated to 60 mmHg for a 20 s stimulation period followed by a 5 min period of rest. Behavioral responses to CRD were collected by a blinded observer in triplicate following a score system based on abdominal withdrawal reflex (AWR). The scores were as follows: 0 = normal behavior without response; 1 = brief head movement at the onset of the stimulus followed by immobility; 2 = contraction of abdominal muscles; 3 = lifting of the abdomen off the platform; 4 = body arching and lifting of pelvic structures [24,25]. 

### 2.6. Determination of Referred Pain

Rats were checked for referred hyperalgesia 7 and 14 days after the last administration of DCA. For this purpose, rats were shaved on the abdomen and then located in individual plastic boxes. Once calm and acclimatized, a series of von Frey filaments (Stoelting Co, Wood Dale, IL, USA) ranging from 8 g down to 1 g were perpendicularly applied to the abdomen. Each filament was tried 5 times for 10 s. If the rat had a positive response (brisk escape), then the filament of the next lower force was applied. The test ended when two filaments were applied without a positive response. 

### 2.7. Intestinal Permeability

Regarding intestinal permeability, 12 days after the last administration of DCA, rats were fasted for 12 h and administered 4000 Da fluorescent dextran–FITC (DX-4000–FITC) (350 mg/kg) by oral gavage. Four hours later, blood was collected from the abdominal aorta and DX-4000–FITC concentration was analyzed in plasma (diluted 1:20) using a Fluorostart fluorescence spectrophotometer (BMG Labtechnologies, Offenburg, Germany). The excitation wavelength was 485 nm and the emission one was 535 nm. FITC–dextran diluted in PBS was used to obtain the standard curves [26].

### 2.8. Gene Expression Analysis

In order to test the impact of the compounds, colonic gene expression of *Muc-3*, *Vegf-α*, *Cox-2* and *Trpv1* were analyzed by RT-qPCR. Total RNA was isolated from colonic samples employing NucleoZOL^®^ (Macherey-Nagel GmbH & Co. KG, Dueren, Germany) following the manufacturer’s protocol. Then, RNA was quantified and reverse transcribed using oligo (dT) primers (Promega, Southampton, UK). Real time quantitative PCR amplification and detection were carried out using 20 ng of cDNA, the KAPA SYBR^®^ FAST qPCR Master Mix (Kapa Biosystems, Wilmington, MA, USA) and specific primers at their annealing temperature (Ta) (Table 1). The values of the housekeeping glyceraldehyde-3-phosphate dehydrogenase (Gapdh) gene were used to normalize mRNA expression. The 2-ΔΔCt method was employed to calculate the mRNA relative quantitation.

### 2.9. Western Blot Analysis

Proteins were extracted from colon samples, their concentration were measured using a BCA Protein Assay Kit (Pierce Biotechnology, Waltham, MA, USA), and run on SDS-PAGE gel (Beyotime Biotechnology, Haimeng, China) and transferred onto a polyvinylidene fluoride membrane (Millipore, Berlington, MA, USA). After 5% milk blocking, the membranes were incubated at 4 °C overnight in the primary antibody: Occludin (E-5) sc-133256 (Santa Cruz Biotechnology, Heidelberg, DE, USA). β-actin (Santa Cruz) at 1:1000 dilution was used as an internal reference. Then, the membranes were incubated with a secondary antibody (Cell Signaling Technology, Danvers, MA, USA) for 2 h at room temperature and exposed to enhanced chemiluminescence for signal intensity quantification (Bio-Rad Laboratories, Madrid, ES, Spain). The obtained images were evaluated using ImageJ Fiji Software [27].

### 2.10. DNA Extraction and Illumina MiSeq Sequencing

Fecal DNA was isolated using the protocol reported by Rodríguez-Nogales et al. [16]. DNA was amplified with primers targeting regions flanking the variable regions 4 through 5 of the bacterial 16 S rRNA gene (V4–5) and explored employing Illumina MiSeq technology. The PCR reactions from the same samples were pooled in one plate, cleaned, and normalized in Invitrogen SequalPrep 96-well Plate kit. The quantified sequences that resulted were completed, quality-filtered, clustered and taxonomically assigned on the basis of 97% similarity level against the RDP (Ribosomal Database Project) [28] with the QIIME2 software package (2021.11 version; https://qiime2.org; California, USA) and “R” statistical software package (version 3.6.0; https://www.r-project.org/) [29]. 

### 2.11. Histology

Colon specimens were fixed in 10% neutral buffered formalin for a minimum of 3 days, dehydrated, embedded in paraffin, and sectioned. Then, histological sections of 5 µm were stained with either hematoxylin and eosin (H&E) or Toluidine Blue (TB). The H&E histological staining was used to assess the inflammatory cell infiltration in the colon. On the other hand, TB staining was applied to visualize mast cells in the connective tissue. Specimens were evaluated independently by 3 individual researchers who were blinded to the experimental procedure. The results were analyzed using ImageJ Fiji Software (version 2.9.0.) [27].

### 2.12. Anxiety Behavior Associated with Visceral Hypersensitivity

Animals were placed in the center of a brightly illuminated arena from the top open-field (OF) for a 10 min trial. The OF arena was a translucent polyethylene box with internal dimensions of 60 cm × 60 cm × 45 cm. Behavioral responses were recorded by a camera above the center of the open-field and connected to a computer. Then, the obtained videos were analyzed by a blinded observer. The time elapsed in central and peripheral zones was recorded using a software designed for this purpose (CPP.OF version 21, Tiselius company, Madrid, Spain). Results were expressed as the percentage of time that the animal spent in the central zone.

The excreted feces were collected and the OF arena was cleaned with ethanol (70%) at the end of each session.

### 2.13. Statistics

In the in vivo and in vitro studies the data are expressed as the mean ± standard error of the mean (SEM) and are representative of three independent experiments. Differences between means were tested for statistical significance using a one-way analysis of variance (ANOVA) followed by the Bonferroni post hoc test. For non-parametric data, median and CI 95% were used and the Kruskal–Wallis test was performed. The von Frey data were registered as areas under the curve (AUC) that were quantified with the trapezoid method. The distribution of force threshold AUC’s was evaluated for normality using the Kolmogorov–Smirnov one-sample test and the data were transformed prior to analysis using the histogram transformation. The AUC data of the five treatment groups were compared using a generalized linear model treatment group as factors.

Microbiome evaluation including α-diversity indices and taxa abundance of the different groups were compared using Kruskal–Wallis test followed by pairwise Mann–Whitney U comparison. Resulting p-values were adjusted by the Bonferroni method. Analysis of α-diversity was carried out on the output normalized data, which were assessed using Mothur. Principal coordinate analysis (PCoA) was accomplished to identify principal coordinates and visualize β-diversity in complex multidimensional data of bacteriomes from different groups of rats. Differences in beta-diversity were tested by permutational multivariate analysis of variance (PERMANOVA). The data are expressed as the mean ± standard error of the mean (SEM). Multiple comparisons between groups were performed using the one-way ANOVA, followed by the Bonferroni post hoc test. 

For the correlation analysis between fold gene expression data and gut microbiome abundance values, the “stat_cor” function of R package was used. We then calculated the Spearman rank correlation coefficients for each group of study and the corresponding p values applying the cor.test() function with a two-sided alternative hypothesis.

All statistical analyses were performed with the GraphPad Prism version 8.1 (GraphPad Software Inc., San Diego, CA, USA), with the statistical significance placed at *p* < 0.05.

## 3. Results and Discussion

IBS is one of the most diagnosed gastrointestinal diseases, with a 10% prevalence in the general population, considering the Rome IV criteria [1]. IBS has a great negative impact on well-being and socioeconomic status, considering that the current treatments frequently show side effects and are not effective [30,31], most likely due to the complicated pathophysiology and the variability of the clinical manifestations of the condition [10]. This has made the patients more prone to self-medicating and using alternative and complementary medicines, which may treat a broader range of symptoms. Among these, probiotics have been proposed for exerting beneficial effects against these conditions [32]. In this study, we have evaluated the effects of the probiotic *L. fermentum* in DCA-induced post-inflammatory IBS in rats. 

### 3.1. L. fermentum Administration Ameliorates the Anxiety-like Behavior in Experimental IBS

Studies conducted in both humans and animals have shown that chronic intestinal diseases are frequently linked to anxiety-like behaviors as well as high levels of stress [9,33]. These behaviors would be the product of changes in corticolimbic areas related to emotion processing [34]. In this study, the open field maze was used to explore if this experimental model of IBS would also be related to anxiety-like behaviors, as well as the impact of the different treatments. In fact, the administration of DCA to rats produced a significant reduction in the time in the central zone compared with Non-IBS rats (Figure 1), which is considered as an evident sign of anxiety. However, those groups treated with *L. fermentum,* rifaximin and gabapentin, significantly spent more time in the central zone, thus indicating an improvement in the state of anxiety associated with the visceral hyperexcitability developed in this model of IBS (*p* < 0.05; Figure 1).

### 3.2. L. fermentum Administration Ameliorates IBS-Associated Visceral Pain

Previous studies have shown that DCA administration to rats for three consecutive days increases visceral hypersensitivity [35], being this the most incapacitating symptom in human IBS. Moreover, this model is characterized by a mild and transient colonic inflammation, with no symptoms of ulceration or epithelial damage and with persistent visceral hyperalgesia. Different methods have been employed to assess this visceral hypersensitivity, which shows some similarities to visceral pain in human IBS [36]. Correspondingly, the results revealed higher visceral hypersensitivity to CRD (pressure applied 60 mm Hg), one and two weeks after DCA administration (IBS group), in comparison with Non-IBS rats (*p* < 0.05; Figure 2a,b). Remarkably, the administration of *L. fermentum* (10^9^ CFU/day) significantly lowered the CRD scores after both periods of time when compared with the IBS control group (*p* < 0.05; Figure 2a,b). Remarkably, the probiotic treatment showed a similar reduction in the CRD values to the treatments with gabapentin or rifaximin (*p* < 0.05; Figure 2a,b), which are frequently prescribed to treat chronic neuropathic pain and visceral hyperalgesia in human IBS [37]. Actually, other probiotics, including lactobacilli and bifidobacteria, have reported beneficial effects on visceral pain in different IBS experimental models [38,39].

### 3.3. L. fermentum Administration Reduces Visceral Hyperalgesia and Allodynia in the DCA Experimental Model

The intracolonic administration of DCA causes moderate intestinal inflammation, associated with somatic hyperalgesia and allodynia [40]. Thus, the referred pain was tested by the response (brisk escape) induced by the application of von Frey filaments in the lower abdomen. The results showed a significantly lower response threshold to pressure in the IBS control group in comparison with the Non-IBS group, revealing the higher visceral hyperalgesia at both time points evaluated (Figure 3a,b). Additionally, the percentages of responses were significantly higher at all the pressure points evaluated, being those more intense after 7 days (Figure 3a) than 14 days later (Figure 3b). The treatment with *L. fermentum* or rifaximin significantly diminished the nociceptive score (from 4 to 26 g), when compared to the IBS group, one and two weeks after the last DCA administration (*p* < 0.05) (Figure 3a,b). Conversely, in the groups treated with gabapentin, although they failed to show significant differences in the nociceptive score in comparison with the IBS control group, a tendency was noticed in the second week and no statistical differences were seen among treated groups. On the other hand, it is important to remark that, despite the fact that other probiotics have been shown to have the potential to modulate visceral hyper sensation and to alleviate the visceral pain responses in animal studies [41,42], this study shows for the first time the capacity of *L. fermentum* to alleviate the increased sensitivity induced by DCA administration in rats.

### 3.4. L. fermentum Administration Ameliorates Gut Inflammatory Status in Experimental IBS

The IBS-associated intestinal inflammatory process plays a pathogenic role in IBS. In fact, several studies have reported gut mucosal inflammation at the microscopic and molecular levels [43]. Moreover, chronic, low-grade, subclinical inflammation has also been thought to perpetuate the symptoms of IBS [35]. Specifically, in this experimental model, the DCA administration also induces a low-grade inflammatory process and an altered immune response [36]. The gene expression profile of different markers, including *Vegf-α, Cox-2* and *Trpv-1*, was significantly increased in the IBS group in comparison with Non-IBS control (Figure 4a–c, respectively). Previous studies have described higher levels of these mediators in IBS patients [44,45,46,47]. Remarkably, the probiotic and rifaximin treatments significantly reduced their expression (Figure 4). These results agree with our previous results regarding visceral pain evaluation. It has been well known that active immunogenic mediators induce activation of COX-2 leading to an increase in the synthesis of prostaglandin E_2_, which contributes to visceral hypersensitivity and diarrhea related to IBS [48]. Here, we can observe that the probiotic and rifaximin treatment reduced the *Cox-2* gene expression, which was associated with the amelioration of the visceral pain and the referred hyperalgesia. Additionally, the increase in TRPV1, a receptor of pain perception, has been reported in gut and primary sensory neurons from animals and IBS patients [49]. It is important to note that different studies have revealed that species from Lactobacillus as well as the rifaximin can improve the visceral hyperalgesia via the TRPV1 channel [50,51]. On the other hand, it is widely recognized that VEGF plays a key role in wound repair and chronic inflammation, being also able to promote vascular permeability, as well as the adhesion of leukocytes to the endothelium and chemotaxis of monocytes [52]. Moreover, VEGF upregulates NF-κB and the production of many proinflammatory cytokines and chemokines [53]. Consequently, a significantly increased gene expression of *Vegf*-*α* was seen in the IBS control group when compared with the Non-IBS control group (Figure 4a), whereas the treatment with rifaximin and the probiotic significantly reduced it. Thus, although it is not well known why these markers are overexpressed, it could be linked to an unbalanced stimulation of the immune response by the intestinal microorganisms, including a mucosal barrier dysfunction, and/or stress-caused activation of the immune system. Consequently, our results sustain that the immunomodulatory properties of *L. fermentum* can contribute to its positive effect.

### 3.5. L. fermentum Administration Reduces Cell Degranulation

As commented above, IBS is more and more considered as a low-grade inflammatory disorder and attention has been centered on the roles of mast cells (MCs) in the intestinal wall. Many other immune cells (T and B cells, eosinophils, monocytes, macrophages) may also participate in the pathophysiology of IBS, however, mast cells have been identified as key connectors of the intestinal mucosa and the nerve fibers of the enteric nervous system [54,55]. MCs are related with major intestinal functions, such as epithelial secretion, epithelial permeability, blood flow, neuroimmune interactions and visceral sensation [56]. In fact, MCs hyperplasia and activation would lead to anomalous gastrointestinal sensitivity, motility and secretion, which in turn promote abdominal pain and/or discomfort, bloating and abnormal intestinal function [57]. In this sense, many therapies targeting mast cells have been explored in IBS patients and have shown good effectiveness to some extent [58,59]. Therefore, and in order to disclose the mechanism involved in the positive effects of *L. fermentum* in the IBS experimental model, the impact of the probiotic was evaluated on the mastocytosis process. The confocal microscopy using blue toluidine staining showed that the number of mast cells in the intestinal mucosa was significantly elevated in the IBS control group compared to Non-IBS rats, being reduced by the different treatments assayed (Figure 5).

Furthermore, the mast cell degranulation process leads to the release of inflammatory mediators that can activate enteric neurons [60]. In fact, previous studies have shown that colonic mast cell infiltration and the mediators released close to the mucosa innervation might promote the abdominal pain perception (severity and frequency) in IBS patients [61]. Consequently, in the present study, the in vitro mast cell degranulation was assessed by determining the β-hexosaminidase release in RBL-2H3 and HMC 1.2 cells. Firstly, the toxicity of *L. fermentum* (10^10^ CFU/mL), gabapentin (5, 10 and 25 µM) or rifaximin (5, 10 and 25 µM) was measured in both cell lines by the MTS assay, and the results suggested that the cell viability was not significantly modified after treatment incubations. However, only *L. fermentum* could significantly decrease the release of the degranulation marker β-hexosaminidase in both cell lines in comparison with non-treated cells (*p* < 0.05). Similarly, the data obtained with C48/80-stimulated RBL-2H3 cells showed that only the incubation with *L. fermentum* significantly inhibited the β-hexosaminidase production in comparison with non-stimulated cells (*p* < 0.05) (Figure 6a). In contrast, when the HMC1.2 cells were analyzed, the results indicated (Figure 6b) that all treatments were capable of reducing the degranulation process, thus showing lower values of β-hexosaminidase production induced by C48/80 when compared with non-treated cells (*p* < 0.0001 all, Figure 6b). Other lactobacilli have been reported to suppress the production of β-Hexosaminidase in vitro [62] and in vivo [63,64]. Interestingly, the C48/80 compound is a mixed polymer of p-methoxy-N-methyl phenylethylamine crosslinked by formaldehyde and broadly employed to stimulate mast cells, thus, many reports have described that C48/80 require stimulation of nerves by mast cell activation. Therefore, these results reveal a new and innovative mechanism of action of the probiotic. Actually, its positive effect can be ascribed to its capacity to modulate the mast cell degranulation through the inhibition of the release of β-Hexosaminidase induced during the inflammatory process.

### 3.6. L. fermentum Supplementation Ameliorated the Altered Intestinal Permeability in Experimental IBS

Several reports have described that a deficient epithelial barrier function leads to aberrant gut permeability, which is linked to a defective immune response and gut dysfunction in IBS patients [65]. In this sense, the IBS group displayed a significant increase in the histological damage of the colon whereas Non-IBS control rats had intact intestinal mucosa, with the villi neatly arranged (Figure 7a,b). After the different treatments, including probiotic administration, an enhancement of the colonic histology was observed, characterized by decreased congestion and edema, as well as inflammatory cell infiltration, regular arrangement of intestinal villi and a smaller villus gap (Figure 7a,b).

It has been described that DCA downregulates the gene expression of numerous pathways associated with cell junctions and improves permeability in a human intestinal barrier model [66,67,68]. Accordingly, a lower expression of the colonic mucin component *Muc-3* was detected in the untreated IBS control group in comparison with Non-IBS rats (*p* < 0.05) (Figure 8a); MUC-3 is an important glycoprotein of the mucus layer found covering the colonic mucosal surface. Actually, the reduction in mucin expression has been previously reported both in this model as well as in other experimental models of IBS [23]. Interestingly, the administration of *L. fermentum*, as well as gabapentin and rifaximin, ameliorated the expression of *Muc-3* (Figure 8a), thus contributing to restoring the damaged gut barrier. Additional experiments showed that *L. fermentum* treatment enhanced the intestinal function by improving permeability, which was evaluated in vivo using FITC-dextran and by the WB determination of occludin expression (Figure 8b). FITC-dextran plasma concentrations in the control IBS rats were higher in comparison with Non-IBS rats, which agrees with an impaired epithelial barrier function seen in this experimental model of IBS. It is interesting to remark that all the treatments significantly lowered FITC-dextran levels in comparison with the IBS control group (*p* < 0.05; Figure 8b). Similar results were obtained when occludin (OCLN) expression was evaluated. It was diminished in the IBS control group in comparison with the Non-IBS group, whereas the treatments were able to increase it (Figure 8c). Curiously, DCA is a secondary bile acid naturally regulated by microbial processes and gut microbial composition. In fact, lactobacilli has been reported as participating in the preservation of the gut barrier by regulating the bile acid metabolization [69]. Consequently, summing up the present results, this probiotic could have a beneficial effect in the IBS condition by improving the intestinal mucosal barrier integrity.

### 3.7. L. fermentum Modulates the Gut Dysbiosis in the DCA Experimental Model in Rats

The IBS pathophysiology comprises alterations in visceral nerve sensitivity, gut permeability and psychological factors. However, more and more evidence suggests the crucial etiopathogenic role played by the intestinal microbiome; in fact, metagenomic studies have reported an altered intestinal microbiome, called dysbiosis, in IBS patients compared with healthy subjects [70,71]. Since previous studies have described the capacity of the probiotic *L. fermentum* to ameliorate gut dysbiosis in different experimental models of disease, the intestinal microbiota composition was evaluated in the present study. The amplicon sequencing results showed a modification in the bacterial architecture of the IBS rats (Figure 9). Specifically, alpha diversity assessed by different indexes (Observed species, Shannon, Simpson, ACE and PD_whole) indicated that the probiotic treatment significantly increased the diversity compared with the untreated IBS group (Figure 9a). Similarly, the beta diversity analyzed by weighted unifrac method revealed a clear separation in the Principal Coordinates Analysis (PCoA) between the Non-IBS and IBS rats, thus indicating an evident disruption of the homeostasis of gut microbiome by the DCA instillation (Figure 9b). Remarkably, *L. fermentum* and gabapentin groups showed a closer distance to the Non-IBS group than to the IBS group (Figure 9b). Consistently, the Venn diagram (prevalence 75%) revealed 34 common OTUs among the Non-IBS and IBS groups, and that the highest number of shared OTUs was found among the Non-IBS rats and the *L. fermentum* group (Figure 9b).

Additionally, the gut microbiota taxa and their abundance were also analyzed. As shown in Figure 10a, the taxonomic composition at the phylum level mainly comprised *Bacillota*, *Bacteroidota* and *Verrucomicrobiota*. The abundance of *Bacillota, Verrucomicrobia* and *Cyanobacteria* was augmented in the IBS group compared with the control group, while *Bacteroidota* and *Pseudomonadota* were reduced (Figure 10a). Notably, the probiotic and gabapentin were capable of ameliorating the modification of gut microbiota composition caused by DCA stimulation (Figure 10a). Moreover, the *Bacillota/Bacteroidota* (named F/B) ratio in the IBS group was significantly greater than in the control rats, and the administration of the probiotic was able to significantly reduce it (Figure 10b). These results confirm previously reported assays in which IBS patients display consistent changes consisting in higher *Bacillota* and lower *Bacteroidota* abundance [70]. In fact, it has also been published that the F/B ratio is increased in IBS patients [71].

At a genus level, the results revealed that the abundance of some bacteria, such as *Prevotellaceae_UCG_001*, *Prevotella_g* and *Collinsella*, was enhanced in the IBS control group when compared to Non-IBS rats (Figure 10d), in agreement with the results previously reported in experimental IBS [72,73]. Of note, the abundance of all of these genera was restored by the probiotic treatment (Figure 10a). Curiously, the overabundance of *Prevotella* has been associated with the induction of visceral hypersensitivity by boosting carbohydrate fermentation and hindering gut mucosal immune function [74,75]. Moreover, the higher abundance of *Collinsella* in IBS rats may display a proinflammatory effect through gas production, thus upholding IBS development [76]. Conversely, different bacterial taxa were diminished in the IBS group in comparison with the Non-IBS group (*Oscillibacter*, *Bacteroides*, *Butyricicoccus* and *Lachnospiraceae*) (Figure 10d), being the abundance of these bacteria partially restored in those groups of rats treated by *L. fermentum* or gabapentin (Figure 10d). Interestingly, the reduction in these bacterial taxa has been also reported in IBS patients [77]. These microorganisms are included in the butyrate-producing bacteria classification and their abundance has been reduced in the IBS group. It is well known that butyrate is a pivotal element for the maintenance of intestinal homeostasis and epithelial integrity, considering that it is the principal energy source for colonocytes. Consequently, the impact of the probiotic administration on these genera can contribute to attenuation of the visceral pain associated with this IBS model. In fact, when we explored the link between *Trpv1* and the abundance of these bacterial genus, the results showed that the IBS group resulted in a positive correlation between *Trpv1* expression and this bacterial abundance, such as *Bacteroides*, *Butyricicoccus*, *Prevotellaceae_UCG_001* and *Collinsella,* compared with the Non-IBS rats (Figure 11). In the *Bacteroides* and *Butyricicoccus* correlation with *Trpv1*, the rise of abundance in the rats treated with gabapentin and the probiotic was not correlated with an increase in the receptor of pain perception (Figure 11a,b). However, when the *Prevotellaceae_UCG_001* and *Collinsella* abundances were correlated with the pain receptor, the treatments maintained lower abundances of both of these bacterial taxa and the expression of the receptor (Figure 11c,d). This relationship occurs reciprocally, since situations of chronic stress, as well as exacerbated anxiety responses, also alter the conformation of the intestinal microbiota [78,79]. This gut–brain relationship could explain how the probiotic treatment improved both intestinal dysbiosis and behavioral disorders [80,81].

## 4. Conclusions

This study revealed the positive effects of *L. fermentum* on an experimental IBS model in rats. *L. fermentum* administration attenuated DCA-induced visceral hyperalgesia and referred pain. Furthermore, the probiotic ameliorated the inflammatory state of the rats, downregulating the expression of pro-inflammatory mediators linked to visceral analgesia and gut epithelial barrier integrity-maintenance. This study suggests a new mechanism of action of *L. fermentum* in the IBS experimental model in rats that involves the serotonin pathway, being the probiotic able to inhibit mast cell degranulation. Moreover, the probiotic exerted beneficial effects through the restoration of gut dysbiosis. Therefore, our data support the treatment with *L. fermentum* as a novel preventative and/or therapeutic strategy against IBS.

## Figures and Tables

**Figure 1 nutrients-15-00024-f001:**
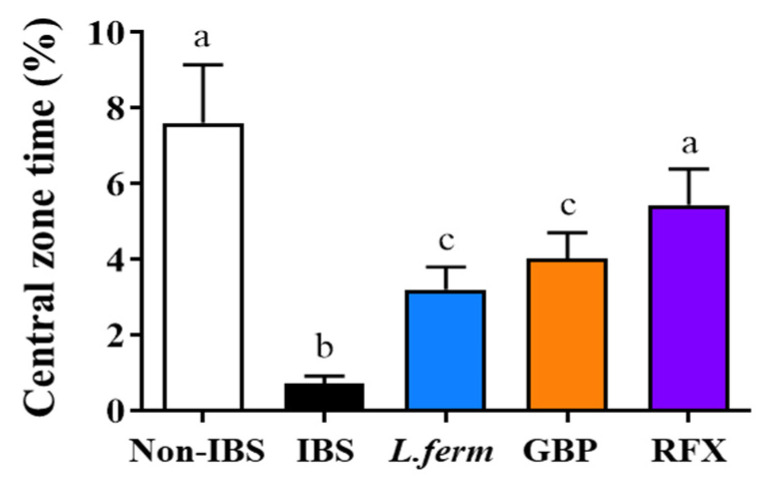
Time spent in central zones of the OF was quantified to assess anxiety-like behavior of DCA vehicle (Non-IBS), 4 mmol/L DCA (IBS), *L. fermentum* (10^9^ CFU) (*L. ferm*), Gabapentin at 70 mg/kg (GBP) and Rifaximin at 150 mg/kg (RFX)-treated rats. Data are expressed as means (triplicate measurements) ± SEM (*n* = 10). Groups with different letters are statistically different (*p* < 0.05).

**Figure 2 nutrients-15-00024-f002:**
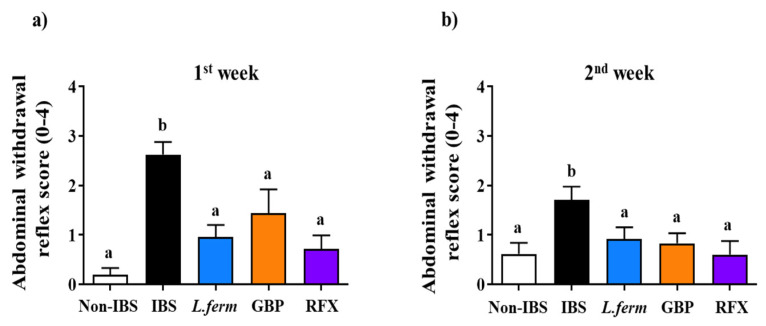
Response to colorectal distension (CRD) was collected using a score system; one (**a**) and two (**b**) weeks after intracolonic administration of DCA vehicle (Non-IBS), 4 mmol/L DCA (IBS), *L. fermentum* (10^9^ CFU) (*L. ferm*), Gabapentin at 70 mg/kg (GBP) and Rifaximin at 150 mg/kg (RFX) treated rats. Data are expressed as means (triplicate measurements) ± SEM (*n* = 10). Groups with different letters statistically differ (*p* < 0.05).

**Figure 3 nutrients-15-00024-f003:**
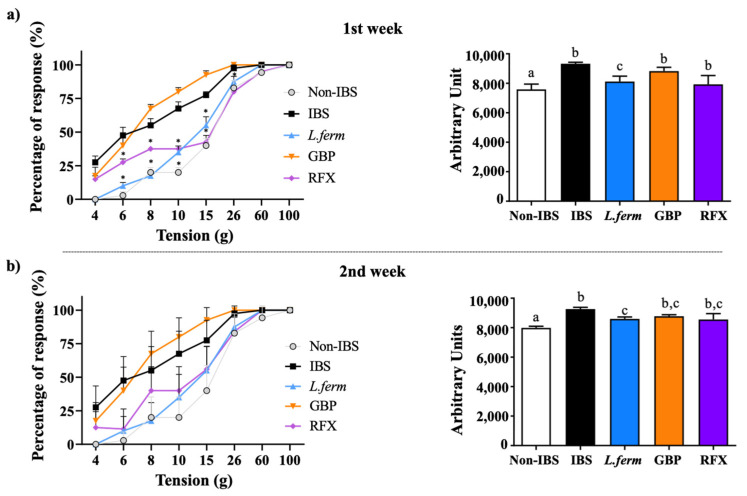
Evaluation of referred hyperalgesia one (**a**) and two (**b**) weeks after DCA instillation. Von Frey filaments (1–8 g) were applied to the abdomen of DCA vehicle (Non-IBS), 4 mmol/L DCA (IBS)-, *L. fermentum* (10^9^ CFU) (*L. ferm*)-, Gabapentin at 70 mg/kg (GBP)- and Rifaximin at 150 mg/kg (RFX)-treated rats. Referred hyperalgesia was calculated considering the percentage of response to filaments (number of animals responding to the filament) and area under the curve (AUC) was determined. Data are expressed as means ± SEM (*n* = 10). * *p* < 0.05 vs. IBS group. Groups with different letters are statistically differ (*p* < 0.05).

**Figure 4 nutrients-15-00024-f004:**
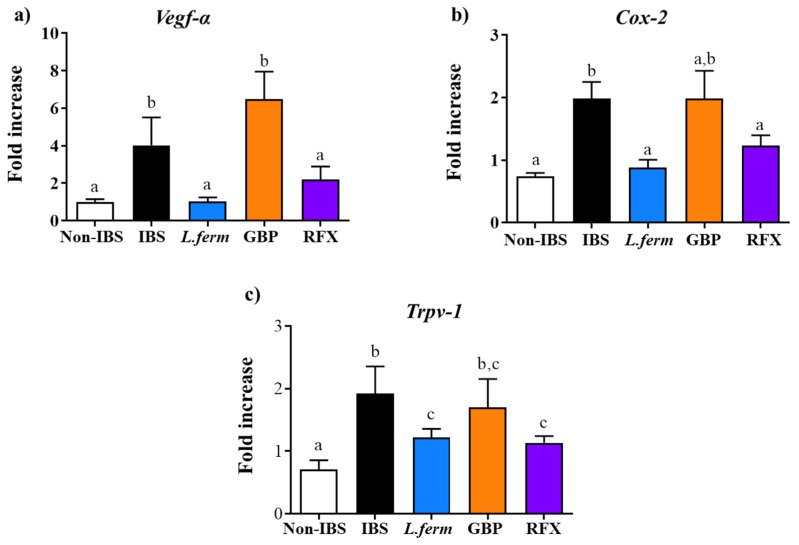
Effect of DCA vehicle (Non-IBS), 4 mmol/L DCA (IBS), *L. fermentum* (10^9^ CFU) (*L. ferm*), Gabapentin at 70 mg/kg (GBP) or Rifaximin at 150 mg/kg (RFX) treatment on colonic gene expression of *Vegf-α* (**a**), *Cox-2* (**b**) and *Trpv-1* (**c**) assessed by real-time qPCR and normalized with the housekeeping gene *Gapdh*. Data are expressed as means ± SEM (*n* = 10). Groups with different letters statistically differ (*p* < 0.05).

**Figure 5 nutrients-15-00024-f005:**
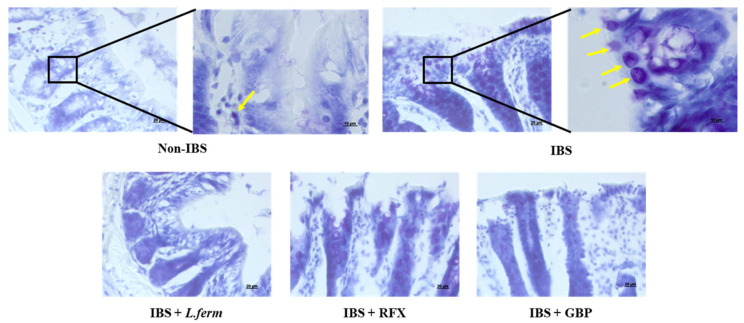
Mastocytosis in colonic tissue in DCA vehicle (Non-IBS), 4 mmol/L DCA (IBS)-, *L. fermentum* (10^9^ CFU) (*L. ferm*)-, Gabapentin at 70 mg/kg (GBP)- and Rifaximin at 150 mg/kg (RFX)- treated rats. Representative distal colon tissue stained with toluidine blue (scale bar = 20 μm). Yellow arrows show mast cells inactivated in Non-IBS tissue and mast cells activated in IBS tissue at 100X.

**Figure 6 nutrients-15-00024-f006:**
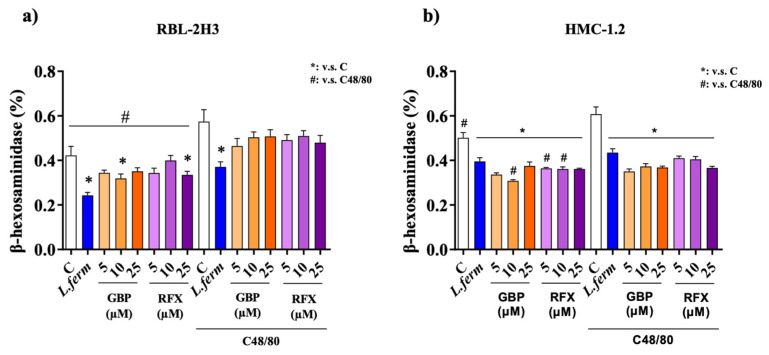
Evaluation of β-hexosaminidase production in RBL-2H3 cells (**a**) and HMC-1.2 cells (**b**). Cell controls (called C) and cells incubated with *L. fermentum* (1 × 10^9^ UFC/mL), Gabapentin (GBP at 5, 10 and 25 μM) or Rifaximin (RFX at 5, 10 and 25 μM) and then stimulated with Compound C48/80 or vehicle. Experiment was performed in triplicate. Data are expressed as means ± S.E.M. Groups with * are statistically different (*p* < 0.05) from the control group (non-stimulated cells). Groups with # statistically differ (*p* < 0.05) from the C48/80 stimulated group.

**Figure 7 nutrients-15-00024-f007:**
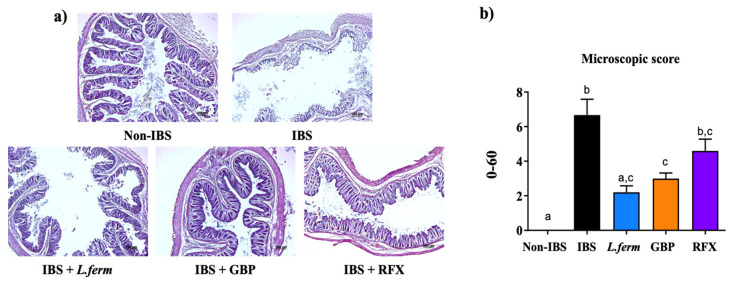
Effects of DCA vehicle (Non-IBS), 4 mmol/L DCA (IBS), *L. fermentum* (10^9^ CFU) (*L. ferm*), Gabapentin at 70 mg/kg (GBP) and Rifaximin at 150 mg/kg (RFX) administration on colonic tissue. (**a**) Representative sections of the distal colon analyzed by hematoxylin and eosin staining (scale bar = 20 μm). (**b**) Histopathological scores of inflammation cell infiltration, depth of lesions, destruction of crypts, width of lesions and crypt damage. Data are expressed as means ± SEM (*n* = 10). Groups with different letters statistically differ (*p* < 0.05).

**Figure 8 nutrients-15-00024-f008:**
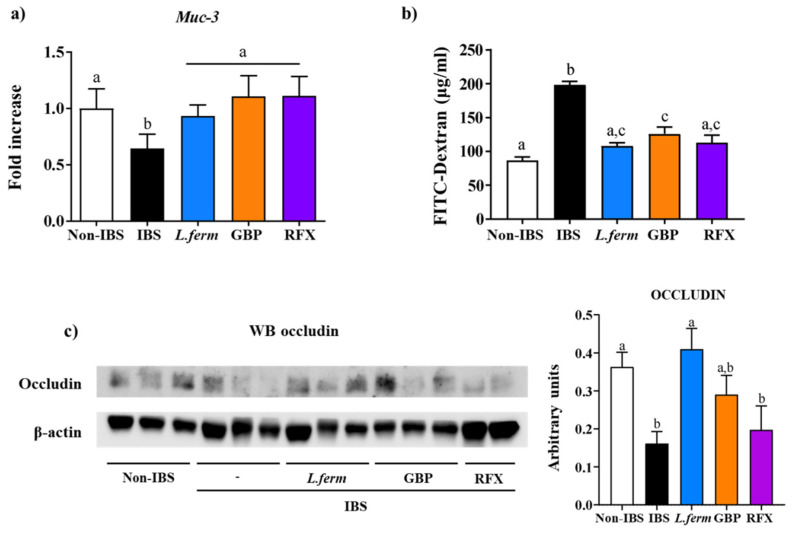
Intestinal permeability evaluation two weeks after DCA vehicle (Non-IBS), 4 mmol/L DCA (IBS), *L. fermentum* (10^9^ CFU) (*L. ferm*), Gabapentin at 70 mg/kg (GBP) or Rifaximin at 150 mg/kg (RFX) administration. (**a**) *Muc-3* gene expression; (**b**) FITC-dextran assay and (**c**) occludin protein expression. Groups with different letters statistically differ (*p* < 0.05).

**Figure 9 nutrients-15-00024-f009:**
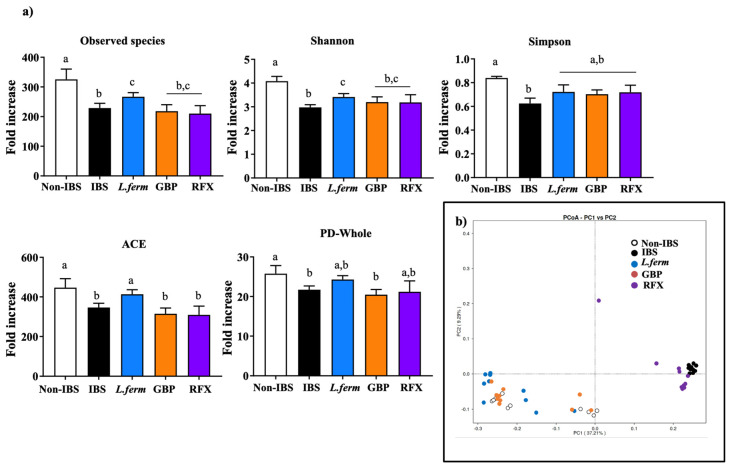
Impact of probiotic administration on microbiome diversity of DCA vehicle (Non-IBS), 4 mmol/L DCA (IBS), *L. fermentum* (10^9^ CFU) (*L. ferm*)-, Gabapentin at 70 mg/kg (GBP)- and Rifaximin at 150 mg/kg (RFX)-treated rats. (**a**) Alpha microbiota diversity indexes determined by Illumina sequencing: Observed species, Shannon and Simpson indexes, ACE and PD-Whole indicator. (**b**) β-diversity calculated by principal coordinate analysis score plot. Data are presented as mean ± SEM. Bars with different letters statistically differ (*p* < 0.05).

**Figure 10 nutrients-15-00024-f010:**
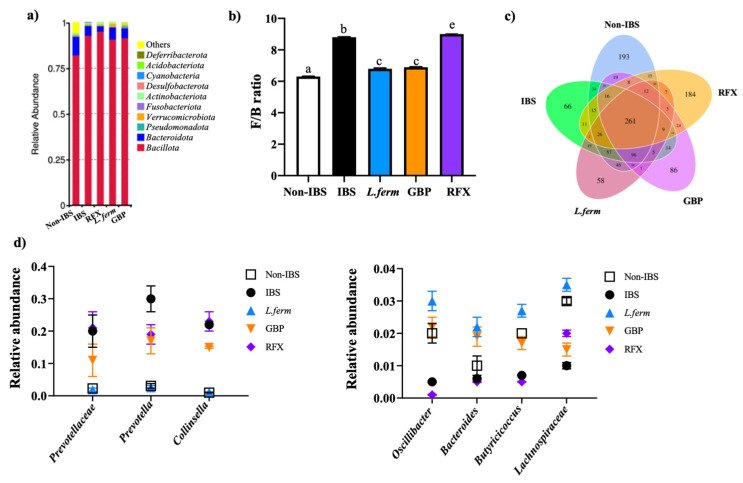
Impact of *L. fermentum* administration on microbiota composition of DCA vehicle (Non-IBS), 4 mmol/L DCA (IBS)-, *L. fermentum* (10^9^ CFU) (*L. ferm*)-, Gabapentin at 70 mg/kg (GBP)- and Rifaximin at 150 mg/kg (RFX)-treated rats. (**a**) Distribution histogram of relative abundance of phylum taxa. (**b**) *Bacillota*/*Bacteroidota* (called F/B) ratio in each experimental group. (**c**) Venn diagram showing the number of OTUs unique and common to each group. (**d**) Taxonomic signatures at genus level. Data are presented as mean ± SEM. Bars with different letters are statistically different (*p* < 0.05).

**Figure 11 nutrients-15-00024-f011:**
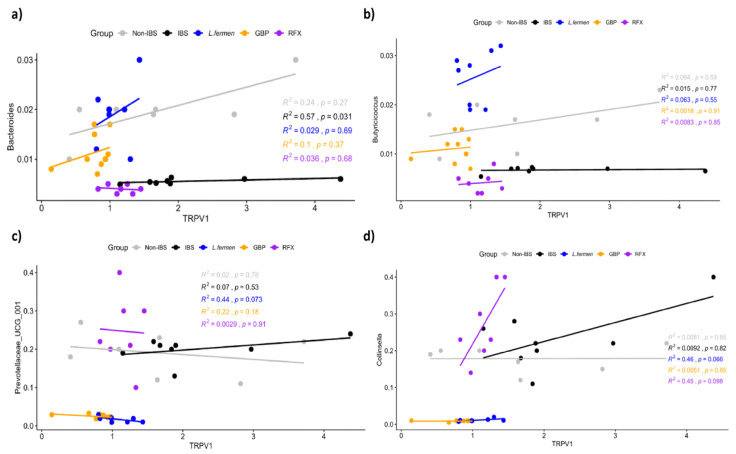
Correlation between *Trpv1* expression and the abundance of (**a**) *Bacteroides* (**b**) *Butyricicoccus* (**c**) *Prevotellaceae_UCG_001* and (**d**) *Collinsella* genus of DCA vehicle (Non-IBS), 4 mmol/L DCA (IBS)-, *L. fermentum* (10^9^ CFU) (*L. ferm*)-, Gabapentin at 70 mg/kg (GBP)- and Rifaximin at 150 mg/kg (RFX)-treated rats.

**Table 1 nutrients-15-00024-t001:** Primer sequences employed for real-time PCR assays.

Gene	Sequence 5′–3′	Annealing Temperature (°C)
*Muc-3*	FW: CACAAAGGCAAGAGTCCAGA RV: ACTGTCCTTGGTGCTGCTGAATG	60
*Cox-2*	FW: TGATGACTGCCCAACTCCCATG RV: AATGTTGAAGGTGTCCGGCAGC	60
*Vegf-α*	FW: CTTCCGAGGGATTCAATATTTC RV: CTCATCTCTCCTATGTGCTG	55
*Trpv1*	FW: AAGAGTTTGTTTGTGGACAG RV: TGTAGTAGAGCATGTTGGTC	56
*Gapdh*	FW: CCATCACCATCTTCCAGGAG RV: CCTGCTTCACCACCTTCTTG	60

## Data Availability

Not applicable.

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
