# Peer review of "Beneficial Effects of Limosilactobacillus fermentum in the DCA Experimental Model of Irritable Bowel Syndrome in Rats"

_nutrients, 2022, doi:10.3390/nu15010024_

Round 1

Author Response

REVIEWER 1

Revision of Nutrients – 2051908

In this paper, the authors analyse the effects of a probiotic bacterial strain (Limosilactobacillus fermentum) in a rat model of IBS, showing a therapeutic effect by reducing several effects of IBS. Furthermore, the authors compare its effect with rifaximin and gabapentin, an antibiotic and an anti-epileptic anticonvulsant, respectively. However, in this paper it appears that gabapentin is used for its effect on peripheral neuropathic pain. Although the study is broad and has taken into account many variables, there are some aspects of the work that need to be addressed.

  1. A first point that the authors should clarify is the experimental design. Although in the introduction they cite a long list of drugs that are used to treat IBS or reduce its symptoms, they do not explain at any point why they choose an antibiotic (rifaximin) and a drug like gabapentin, which is used to treat peripheral neuropathic pain. Although I can understand the comparison of the probiotic with the antibiotic, I do not understand the comparison with gabapentin as the mechanisms of action of the two are far apart. Furthermore, nowhere do they state that rifaximin is an antibiotic and that gabapentin is used to treat peripheral neuropathic pain. I think they should explain what they are in the material and methods section, as well as indicate the objective of their comparison with probiotic bacteria at the end of the introduction section, in the objective.

Response: We thank the reviewer for the comments and suggestions. Accordingly, in the introduction section, specifically in the objective, we have explained the rationale for the use of these two treatments as control groups.  As now reported in the text, both treatments have been clinically tested for different characterized symptoms in IBS patients.

  1. There is inconsistency regarding the number of animals used. On line 113 the authors indicated that the animals are housed in groups of 5 per cage. Whereas in line 115 they stated that there was n=8. In the caption of some figures (not of all), the authors indicated an n of 10 animals per group. The authors should clarify the number of animals used in each experiment and state it in the captions of the corresponding figures.

Response: We are sorry for the mistake. The manuscript has been revised and corrected. 10 rats/group housed in cages (5 rats/cage) have been used.

  1. Regarding the statistics, the test applied to the correlations shown in figure 11 is missing. Moreover, the correlations are not made group by group, but taking into account all the animals together.

Response: We performed correlation analysis between gene expression data and gut microbiome abundance data for each group of study. We apologize if this analysis was not well described. Consequently, a paragraph including a more detailed analysis description of the statistical analysis has been included in material and methods. In fact, Spearman correlation was used for this analysis as it performs better with normalized counts (gene expression) as well as compositional data (microbiome relative abundance) compared to other metrics, such as Pearson correlation [1]. 

  1. Figure 1 does not show how the results are expressed and the number of animals used.

Response: We apologize for this mistake and have now included the information in figure 1.

  1. In Figure 3, the authors indicate that they express the hyperalgesia referred to as percentage response to the filaments and that the area under the curve (AUC) was calculated. But there is no explanation as to how the percentage was calculated and what it is expressed for. Furthermore, the comparison of the area under the curve is also not indicated in the Statistics section of the Materials and methods section. Furthermore, the titles of the graphs on the right are missing. In the right-hand chart in panel b) the y-axis title is also missing.

Response: We thank the reviewer comments. We have modified the figure legend. To determine the percentage response to the filaments, we divided the number of animals with brisk escape by the total number of animals and then multiplied the resultant by 100. This calculation was made for each filament applied. Regarding the statistical method used in the comparison of the AUC, we have included in the material and methods section a more detailed information.

Moreover, the figure 3 has been corrected and the missing titles have been included.

  1. Figure 7 shows histological images of the colon of rats from the different experimental groups. In lines 393 and 394, they indicate that there is significant histological damage, but the authors have not determined the histopathological index, nor have they compared it between the groups to see if the damage is significantly more or less. To improve both the analysis of the results and to strengthen the conclusions, they should quantify the histopathological index of the different animals and analyse them statistically. Moreover, in the figure legend, the authors should indicate that these images are representative images.

Response:  A new figure with the microscope score has been included as Figure 7b.

  1. Figure 8 shows western blot images of occludin abundance in the colon of rats from the different experimental groups. In lines 419 and 420, they indicate that it is reduced in the IBS animals and that the treatments increase it, but the authors have not quantified the density of the bands, nor have they compared it between the groups to know if it is really reduced or not once it has been normalised with the reference protein and the damage is significantly higher or lower. Moreover, the authors indicated in the Western blot analysis, in the Section of Material and Methods, that “Western blot images were analysed using ImageJ Fiji software”. To improve both the analysis of the results and to strengthen the conclusions, they should quantify the band density of all animals and analyse them statistically.

Response: We apologize for this mistake because the occludin quantification was not included in the figure 8. We have now included in the figure 8 the occludin quantification.

  1. The authors do not discuss why gabapentin does not reduce the hyperalgesia referred to (Figure 3), when it is one of its therapeutic targets. I think they could discuss why this happens and the possible pathophysiological mechanisms that may differentiate between the effect of the probiotic bacteria and the drug used.

Response:  We appreciate the reviewer’s comment. As shown in Figure 3, when the hyperalgesia referred was evaluated using von Frey filament test, the gabapentin treatment did not reduce it when it was determined in the first week. However, in the second determination, gabapentin showed a trend to reduce hyperalgesia, and no statistical differences were observed with the other treated groups. Moreover, it is important to highlight that the visceral pain evaluated by colorectal distension revealed that gabapentin treatment significantly reduced it. In this context, analgesic effects on peripheral nociception in rats have been previously reported to injectable gabapentin [2]. Therefore, our results could be probably associated with different factors such as the bioavailability of the drug, the dose, and the route of administration. However, the focus of this study was to compare the effect of the probiotic with gabapentin, a drug used in the treatment of human IBS. Therefore, we use the oral route, which is how it is administered in humans.

Minor concerns

  1. They should indicate what is the Compound 48/80 and what it is used for. They should also indicate in the Material and methods section where it was obtained, as well as indicate who supplies the drugs used.

Response: Following the reviewer's suggestions, we have explained the use of C48/80, as well as the supplier. 

  1. In the caption to figure 5, the authors should indicate that representative images of the different experimental groups are shown.

Response: According to the reviewer suggestion, we have indicated that representative images were used.

  1. In figure 9b), why not use the same colours as in the rest of the figures to indicate the different groups? I think that using the same colours as in the rest of the work would make it easier for the reader to understand.

Response: We apologize for the inconvenience and accordingly we have modified the figure 9b.

  1. In Table 1, the primer sequence of the Gapdh is missing.

Response: We apologize for the mistake, the Gapdh sequence has been included in table 1.

  1. There are some spaces throughout the manuscript. They should be corrected.

Response: Accordingly, the manuscript has been revised.

References

  1. Weiss, S., et al., Correlation detection strategies in microbial data sets vary widely in sensitivity and precision. Isme j, 2016. 10(7): p. 1669-81.
  2. SM, O.M., et al., The effects of gabapentin in two animal models of co-morbid anxiety and visceral hypersensitivity. Eur J Pharmacol, 2011. 667(1-3): p. 169-74.

Reviewer 2 Report

It is a well-designed study on IBS in animal model. The aim is clear. Methods were described properly. Results and discussion are well written. The results are promising for humans.

Author Response

It is a well-designed study on IBS in animal model. The aim is clear. Methods were described properly. Results and discussion are well written. The results are promising for humans.

Response: We greatly appreciate the positive feedback and comments made by the reviewer.

Author Response

Materials and Methods

  1. Line 88, (MTS) should be (MTT).

Response: The abbreviation is correct. However, a mistake was made in the protocol of MTS assay. Accordingly, the manuscript has been modified.

  1. Please give more detail about the medium used for cell culture.

Response: The medium composition has been included in the manuscript, section materials and methods.

  1. Line 101, released of the b-hexosaminidase….., should be released bhexosaminidase

Response: We apologize for the mistake and the sentence has been rephrased.

  1. Line 113-115, the rats were housed 5 rats per cage, but only 8 rats per groups, please confirm.

Response: The mistake has been corrected. This study was performed using 10 animals per group of which 5 rats were housed per cage.

  1. The administration of DCA was described in different terms (administration, injection, installation) in the text, it easily makes confusion. Please unify.

Response: We appreciate the reviewer suggestion. Consequently, the manuscript has been revised and the terms have been unified.

Results and discussion

  1. The superscripts used to indicate the significant difference between groups in all figures should be labeled according to the increasing or decreasing values.

Response: We thank the reviewer's comment. The superscripts used in the figures indicate the significant differences between groups. Thus, different letters indicate significant differences between groups (p<0.05).

  1. Please point out the sample number used in each experiment.

Response: The sample number used in each experiment has been included in the text and in the figure legend.

  1. AUC of Figure3 seems doubled the real values, please re-check.

Response: We have checked the figure 3, and the values are correct. In order to clarify this analysis,  we have included a more detailed explanation about the AUC calculation in the materials section.

  1. Line 317, Lactobacillus should be in italic form.

Response: It has been corrected.

  1. In materials and methods, 8 rats in each group is described, but in Figure 2, 3, and 4, the sample number are 10, how could they be?

Response: As commented above, we apologize for this mistake. This study was performed using 10 animals per group of which 5 rats were housed per cage.

  1. According to 10a’s data, the abundance of Bacillota are larger than 0.75 and Bacteroidota are less than 0.125, the F/B ratio should be larger than 6. But the F/B ratio in Figure 10b are all less than 1, but Please confirm.

Response: We apologize for the mistake. The figure has been corrected.

  1. In Figure 10d, Prevotellaceae and Lachnospiraceae are both Families, not genera, please correct.

Response: The mistake has been corrected.

  1. Line 501, Trpv1 should be in italic form.

Response: The mistake has been corrected.

Round 2

Reviewer 1 Report

OK

Author Response

The manuscript has been revised